# Real-IKEA: Simulating What Robots Will Really See and Touch

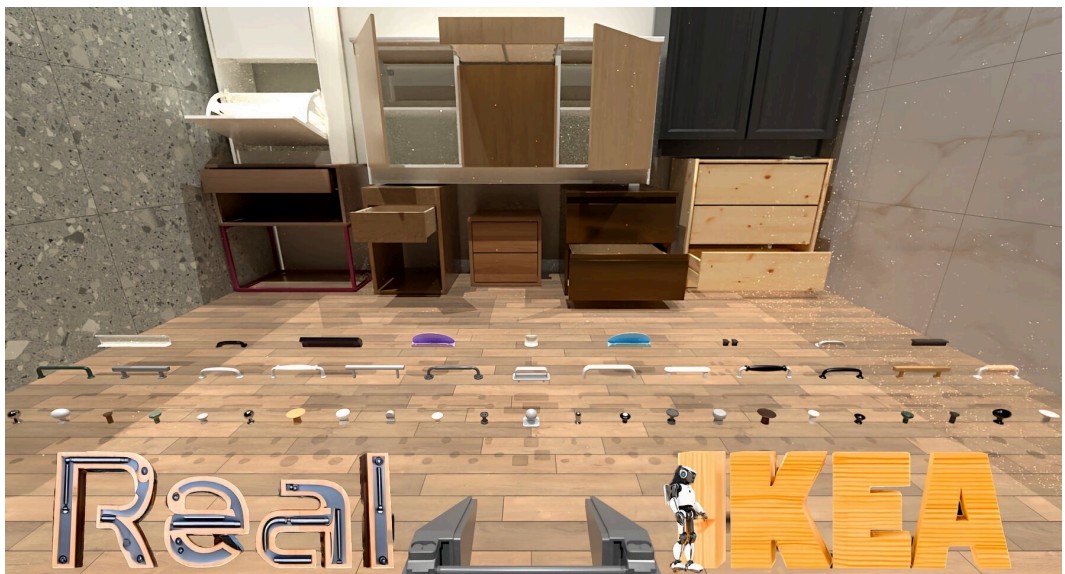

Figure 1: **Overview of Real-IKEA.** Real-IKEA provides a new dataset and simulation framework for contact-rich articulated object manipulation. The assets are sourced from real IKEA furniture and each modeled with precise convex decomposition. The framework supports both realistic teleoperation and high-fidelity rendering, enabling systematic studies of contact-rich manipulation.

## Abstract

Robotic manipulation has greatly benefited from simulated data, yet in contact-rich tasks policies often fail to transfer. We trace this sim-to-real gap to three sources: **object assets**, **physical realism** and **visual fidelity**. We emphasize *accuracy* along all three axes—precise meshes and collisions, calibrated friction and hinge resistance, and visually realistic observations—and present **Real-IKEA**, a dataset and simulation framework designed with accuracy as a first-class goal. At scale, Real-IKEA provides **1,079** articulated asset configurations, created by combining real IKEA furniture bases with a curated library of **83** authentic IKEA handles and knobs. For contact-geometry accuracy, we introduce a bidirectional surface-deviation metric ($E_{Q \to P}$, $E_{P \to Q}$) that quantifies collision meshes against the visual mesh. For dynamics accuracy, we establish resistance-calibrated benchmarks that vary damping and friction. To narrow the vision gap, we pair real-time teleoperation with offline high-fidelity re-rendering and quantify alignment via FID/EMD across multiple encoders. Extensive comparisons show that Real-IKEA yields more realistic asset structure, more accurate physical interactions, and visuals more closely aligned with real data, enabling policies to exploit geometry and torque rather than rely on friction-only pulling. This accuracy-centric design, coupled with large scale, enables the scalable collection of reliable manipulation data and more robust sim-to-real transfer.

# 1 INTRODUCTION

Learning manipulation policies for robots has long relied on simulation as a scalable and cost-effective source of data (Mo et al., 2019; Xiang et al., 2020; Gu et al., 2023; Li et al., 2024c; Wang et al., 2025b). The paradigm of the *data pyramid* illustrates this strategy: massive amounts of synthetic data form the base, supplemented by smaller amounts of curated real-world data at higher tiers. This approach has proven effective in locomotion, navigation, and other domains (Ye et al., 2025; Rudin et al., 2022; Sferrazza et al., 2024; Tan et al., 2018). Yet in *contact-rich manipulation tasks*, synthetic data often falls short. Policies trained on low-quality simulated data rarely transfer reliably to the real world (Blanco-Mulero et al., 2024; Jaunet et al., 2021; Yardi et al., 2025). The root cause is not the learning algorithm itself, but rather the systematic unrealism of environment modeling, physical simulation, and visual rendering in existing simulators.

To address this, the community has invested in collecting real-world teleoperation data. Despite its scarcity, such data provides high-quality demonstrations for imitation learning (Zhao et al., 2023; Cheng et al., 2024b; Fu et al., 2024; Iyer et al., 2024; Heng et al., 2025). However, scaling real-world data collection is costly, time-consuming, and limited in diversity. As a result, policies tend to be tied to specific object instances or simple actions, and generalization remains weak. What we ultimately seek is for robots to rapidly acquire the ability to manipulate diverse objects across varied environments. This motivates an urgent question: *Can we improve the quality of simulation data itself, so that abundant synthetic data can meaningfully support real-world deployment?*

We identify three major limitations of current simulation data for manipulation:

**Unrealistic object assets.** Prior datasets such as PartNet-Mobility (Xiang et al., 2020) include a large variety of assets, but these are synthetically generated and often diverge from real-world distributions. For instance, many hinge-based cabinets appear in PartNet-Mobility, but with over-simplified handles and stylistic biases toward traditional furniture. This mismatch introduces out-of-distribution (OOD) risks when deploying to modern environments.

**Inaccurate physical interactions.** Mainstream physics engines and simulators (Todorov et al., 2012; Makoviychuk et al., 2021; Mittal et al., 2023) rely on convex decomposition for collision handling. Without sufficiently detailed decompositions, handles may lack holes or fine-grained geometry, making certain real-world strategies—such as hooking a handle—impossible to reproduce in simulation. This prevents learning realistic and potentially superior manipulation strategies.

**Low-fidelity visual observations.** Unlike locomotion, where proprioception dominates (Serifi et al., 2024; Cheng et al., 2024a; He et al., 2025), manipulation policies heavily depend on external visual input. Yet simulated RGB renderings differ significantly from real camera observations (Li et al., 2024b), creating a visual domain gap that undermines transfer.

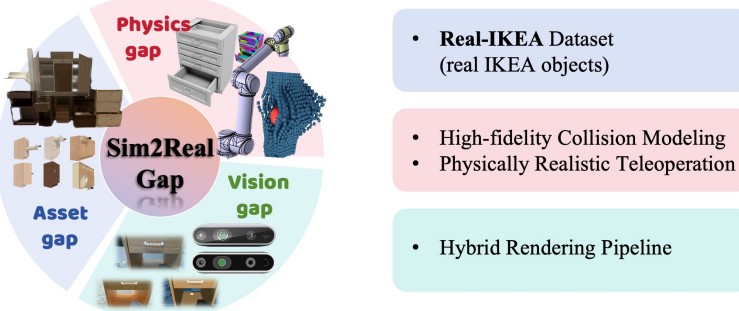

Figure 2: Three key limitations in Sim2Real transfer: **asset gap**, **physics gap**, and **vision gap**. Real-IKEA addresses these gaps to enable more reliable simulation and policy learning.

To overcome these challenges, we introduce **Real-IKEA**, a high-quality dataset and simulation framework designed to improve manipulation data across all three dimensions. First, we build assets directly from real IKEA furniture, using official designs and Real2Sim meshes to ensure realistic structures and appearances. This reduces the gap between simulated and real object distributions. Second, we curate a large set of handles and knobs—83 instances across 47 unique designs—with

precise convex decompositions, enabling physically faithful interactions and the emergence of realistic strategies. Third, we adopt a hybrid rendering pipeline: while teleoperation runs in real time, all state changes are recorded and re-rendered offline using high-quality rendering, yielding photorealistic visual data for learning.

Empirically, we validate Real-IKEA along all dimensions. Physics-based evaluations show more realistic interactions. Representation learning analyses reveal that features from our offline-rendered observations align more closely with real camera data. Finally, we implement a teleoperation framework within Real-IKEA and demonstrate the shortcomings of existing manipulation policies under realistic conditions. Together, these enable scalable collection of higher-quality manipulation data in simulation while preserving sim-to-real transferability. We believe Real-IKEA marks a crucial step toward scalable and realistic training environments for next-generation robotic manipulation. **Our main contributions are:**

- **Real-IKEA dataset**. We source articulated object assets from real IKEA furniture: base cabinet units combinatorially paired with a curated library of 83 reusable handles/knobs, yielding $1,079$ realistic assets with consistent real-world distributions.

- **High-fidelity physical interaction modeling and teleoperation**. Precise convex decompositions for collision, resistance-calibrated joints (damping/friction), and a teleoperation stack that enables collecting contact-rich demonstrations.

- **Hybrid rendering pipeline**. Real-time rendering for data collection combined with offline high-quality re-rendering, quantitatively narrowing the visual gap.

- **Comprehensive evaluation and benchmark**. Metrics for collision accuracy and visual fidelity, resistance-calibrated tasks across handle types, and analyses that expose limitations of friction-dominated policies—establishing a robust benchmark for contact-rich manipulation.

## 2 RELATED WORKS

### 2.1 ARTICULATED OBJECT SIMULATION ASSETS AND FRAMEWORKS

Existing simulation asset libraries and frameworks have made important progress in enabling manipulation research. PartNet-Mobility (Xiang et al., 2020; Mo et al., 2019) contains a large number of articulated objects, but both its visual and collision meshes are relatively coarse, limiting realism in contact-rich tasks. Adamanip (Wang et al., 2025a) provides finer visual meshes and cleverly reuses interactive components to scale the number of assets. However, its collision meshes remain coarse, and the diversity of truly interactive parts is still limited. The ManiSkill series (Tao et al., 2025; Gu et al., 2023) introduces multi-task, multi-object benchmarks, yet most of its assets are simplified CAD models or synthetic geometries. Other embodied AI benchmarks (Li et al., 2024a; Srivastava et al., 2021; Szot et al., 2022; Savva et al., 2019; Kol, 2017) focus primarily on navigation and reasoning, with insufficient contact modeling. Overall, existing assets often lack realistic furniture appearance and precise contact geometry, which contributes to the sim-to-real gap, especially for contact-rich manipulation. This gap helps explain why few products today can reliably manipulate articulated objects in real indoor environments.

### 2.2 ARTICULATED OBJECT MANIPULATION POLICY LEARNING

Recent progress has been made in policy learning for articulated object manipulation. One line of work uses affordance prediction on point clouds to capture the influence of hinge structures on object motion (Mo et al., 2021). Other approaches leverage foundation or segmentation models to combine component localization with inference of manipulation mechanisms (Zhang et al., 2025; Huang et al., 2024). Reinforcement learning-based approaches also enable manipulation through exploration (Zakka et al., 2025; Nguyen & La, 2019). However, these methods all simplify the task to varying degrees. Affordance-based methods rely on predefined actions, lack scalability, and show low success rates with simple pulling motions. Foundation and segmentation model-based approaches are typically restricted to cases with simple end-effector postures. RL-based methods often operate in overly simple environments or excessively large exploration spaces. As a result, these strategies fail to deliver robust real-world performance. They frequently assume simplified collision

models, ignoring friction and resistance in articulated joints. In practice, humans exploit postures that maximize force application. Ignoring resistance prevents learned strategies from adapting to different components, such as adjusting the end-effector pose based on handle or knob geometry. Large grasping models Murali et al. (2025); Ye et al. (2025) have shown success in simple grasp tasks, but are not designed for articulated interactions and their performance under joint resistance remains unclear. Thus, a realistic simulation environment is needed to develop strategies that address these real-world challenges.

## 2.3 Physical Realism and Visual Fidelity in Contact-Rich Manipulation

High-quality assets are essential for reliable manipulation research. On the physical side, collision meshes should be generated through convex decomposition methods such as COACD (Wei et al., 2022). Coarse approximations fail to capture fine-grained handles, grooves, holes, and curved structures. While some works adjust control parameters to better match real-world dynamics, few explicitly model interactive parts with high accuracy. Most efforts emphasize policy robustness rather than asset fidelity. In terms of physical consistency, little research systematically compares the errors between simulated collisions and real contacts of key articulated components. On the visual side, some works employ background replacement or green-screen augmentation to improve realism (Li et al., 2024b), but these methods require fixed environments and cannot generalize well. Therefore, improving physical contact modeling and visual alignment remains an open challenge.

## 3 Real-IKEA Dataset and Simulation Framework

Real-IKEA aims to enhance simulation data quality across the three key dimensions outlined in Figure 2. Its assets are sourced directly from real IKEA products, ensuring high consistency with the physical world. More sophisticated collision and physics modeling enables contact-rich operations that would otherwise be infeasible to simulate. A hybrid rendering pipeline further improves visual quality when collecting teleoperation data in simulation.

| Environment | Reusable Interactive Parts | Accurate Collisions (Decomp.) | Configurable Joint Resistance (Damp./Fric.) | Real-Object Digital Twins |
|---|---|---|---|---|
| SAPIEN | ✗ | ✗ | ✗ | ✗ |
| AdaManip | ✓ | ✗ | ✗ | ✗ |
| UniDoorManip | ✓ | ✗ | ✗ | ✗ |
| **Real-IKEA** | ✓ | ✓ | ✓ | ✓ |

Table 1: Feature comparison across articulated-object datasets and simulation environments.

## 3.1 Real-IKEA Asset Dataset

A central question in building an articulated object dataset is: how are such objects actually designed in the real world? Rather than synthesizing arbitrary geometries, we turn to IKEA, whose products provide a globally standardized yet widely deployed set of household furniture. We adopt IKEA cabinets as the canonical base units and ensure coverage of all common joint mechanisms. This choice is motivated not only by their prevalence in daily life, but also by IKEA's design philosophy, which emphasizes modularity and reusability—precisely the properties that facilitate generalization in manipulation learning.

A recurring challenge in prior datasets is the trade-off between diversity and reusability of operable parts. For example, SAPIEN emphasizes cabinet shape variation but largely overlooks systematic modeling of reusable handles and knobs. Adamanip, in contrast, leverages part reuse to scale asset count, but its operable components remain morphologically limited. In practice, generalizable policies must be trained on parts that are diverse, reusable, and realistically modeled. IKEA's modular construction naturally offers this advantage: handles and knobs are manufactured as standardized, interchangeable units, mounted onto base cabinets to create a wide range of configurations. Following this principle, we curated a complete set of 83 IKEA handles and knobs, systematically

combining them with base cabinets to yield **1,079** articulated assets. A cross-environment comparison in Table 1 further situates Real-IKEA among existing simulators, highlighting that it uniquely combines reusable interactive parts, accurate collision modeling via convex decomposition, configurable joint resistance, and real-object digital twins.

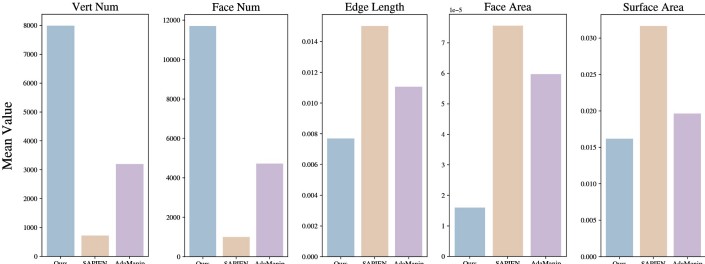

Figure 3: Quality of the Real-IKEA dataset. Our meshes achieve the highest vertex counts, shortest edge lengths, most faces, and the smallest average face areas.

This construction provides a unique advantage: it enables principled study of contact-rich manipulation. While it is widely recognized that simple strategies—e.g., grasping with fixed orientations or relying on predefined primitives—are insufficient, the field has lacked assets with sufficiently accurate contact modeling to move beyond such abstractions. Real-IKEA closes this gap.

Quantitatively, we compare against assets from the widely used PartNet-Mobility dataset and Adamanip, focusing on interactable components. Figure 3 indicates a significantly finer level of geometric detail in our meshes. Furthermore, unlike PartNet-Mobility, whose object scales are inconsistent with real-world dimensions, Real-IKEA assets are faithfully aligned with physical measurements, avoiding distortions in contact geometry. Taken together, Real-IKEA provides a large-scale, physically grounded, and morphologically diverse library of articulated objects. Its combination of high-fidelity geometry and real-world consistency makes it uniquely suited as a foundation for learning and evaluating contact-rich manipulation policies.

## 3.2 PHYSICAL INTERACTION

While high-quality visual meshes improve the realism of simulated assets, they do not automatically translate into realistic physical interactions between a robot's end-effector and articulated objects. A key limitation arises from the fact that most simulation platforms only support convex collision meshes: non-convex geometries must therefore be approximated via convex decompositions. In many prior datasets, the collision mesh is either directly reused from the visual mesh or crudely approximated by convex hulls. Despite its ubiquity, the magnitude of this approximation error—and its impact on contact-rich manipulation—has not been systematically studied.

To address this gap, we reconstruct collision meshes using the COACD (Wei et al., 2022) algorithm, which produces high-fidelity convex decompositions better aligned with the true visual geometry. Beyond reconstruction, we also propose a quantitative metric to evaluate collision mesh accuracy in contact-rich tasks. We treat the visual mesh as ground-truth geometry and uniformly sample dense surface points on both the visual mesh (*standard shell*) and the collision mesh (*collision shell*). For each point $q$ on the collision shell $Q$, we compute its nearest-neighbor distance to the standard shell $P$. This measures the outward deviation of the collision shell relative to the true geometry. The overall deviation is averaged across all collision-shell points:

$$E_{Q \to P} = \frac{1}{|Q|} \sum_{q \in Q} \operatorname{dist}(q, P), \qquad \operatorname{dist}(q, P) := \min_{p \in P} \|q - p\|_2.$$

Symmetrically, we compute $E_{P \to Q}$ by swapping the roles of $P$ and $Q$, yielding a bidirectional measure of geometric discrepancy. Figures 4 and 5 illustrate this evaluation. Without reconstruction, baseline collision meshes show large deviations, especially in handles, grooves, and curved support structures—the very regions most critical for grasp stability. Our COACD-based modeling substantially reduces average deviation and enables fine-grained error visualization via heatmaps. This analysis further reveals that conventional approximations can completely miss holes or curved

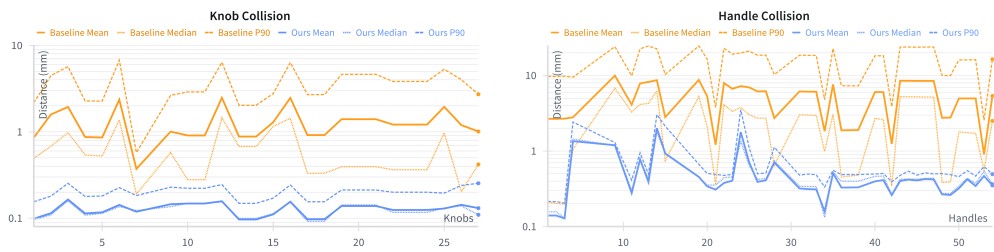

Figure 4: Evaluation of physical interaction fidelity. We report $E_{Q \to P}$ and $E_{Q' \to P}$ for each Real-IKEA interactive asset ($Q'$ denotes the baseline collision shell without our processing). Our reconstructed meshes achieve significantly higher accuracy compared to the baseline.

surfaces, rendering them unsuitable for studying contact-rich manipulation. By contrast, our reconstructed meshes preserve these critical features, making simulation outcomes far more consistent with physical reality.

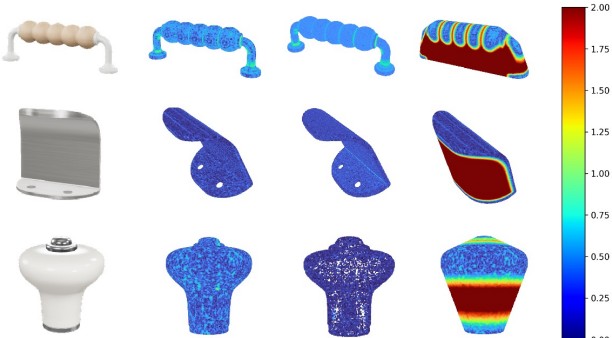

Figure 5: Heatmap visualization of collision errors. The left column shows $H_{P \to Q}$, the middle column shows $H_{Q \to P}$, and the right column shows $H_{Q' \to P}$.

## 3.3 VISUAL RENDERING

A major contributor to the sim-to-real gap lies in discrepancies in visual rendering. Although Isaac Sim provides one of the most photorealistic rendering pipelines among robotics simulators, its physical dynamics are less accurate than those of MuJoCo. For this reason, our contact-rich manipulation environments are built on MuJoCo. Nevertheless, simulator renderings inevitably diverge from real-world camera inputs, both because computational rendering cannot fully capture physical light transport and because real-time constraints force a trade-off in fidelity.

To mitigate this issue, we adopt a hybrid rendering pipeline. During teleoperation, lightweight real-time rendering is used for efficient data collection, while trajectories are subsequently re-rendered offline with a high-quality renderer (e.g., Mitsuba3). This strategy improves the visual realism of collected data without sacrificing efficiency, thereby narrowing the visual sim-to-real gap.

To evaluate its effectiveness, we compare embeddings of simulator renderings, enhanced renderings, and real-world camera frames using multiple pretrained visual encoders (ResNet-50, ResNet-18, CLIP ViT-B/32). Distributional similarity is quantified using Fréchet Inception Distance (FID) and Earth Mover's Distance (EMD):

$$D_{\mathrm{FID}}(\mathcal{E}_1, \mathcal{E}_2) = \|\mu_1 - \mu_2\|_2^2 + \mathrm{Tr}\big(\Sigma_1 + \Sigma_2 - 2(\Sigma_1 \Sigma_2)^{1/2}\big),$$

$$D_{\mathrm{EMD}}(\mathcal{E}_1, \mathcal{E}_2) = \inf_{\gamma \in \Gamma(\mathcal{E}_1, \mathcal{E}_2)} \int \|x - y\| \, d\gamma(x, y),$$

where $(\mu_i, \Sigma_i)$ are the mean and covariance of embeddings $\mathcal{E}_i$, and $\Gamma(\mathcal{E}_1, \mathcal{E}_2)$ denotes the set of valid couplings. Lower values of $D_{\mathrm{FID}}$ and $D_{\mathrm{EMD}}$ indicate smaller visual gaps.

As shown in Table 2, our hybrid rendering pipeline substantially reduces both FID and EMD compared to raw simulator outputs. PCA and t-SNE visualizations (Figure 6) further demonstrate that enhanced renderings form distributions closer to real-world data, validating the effectiveness of our approach.

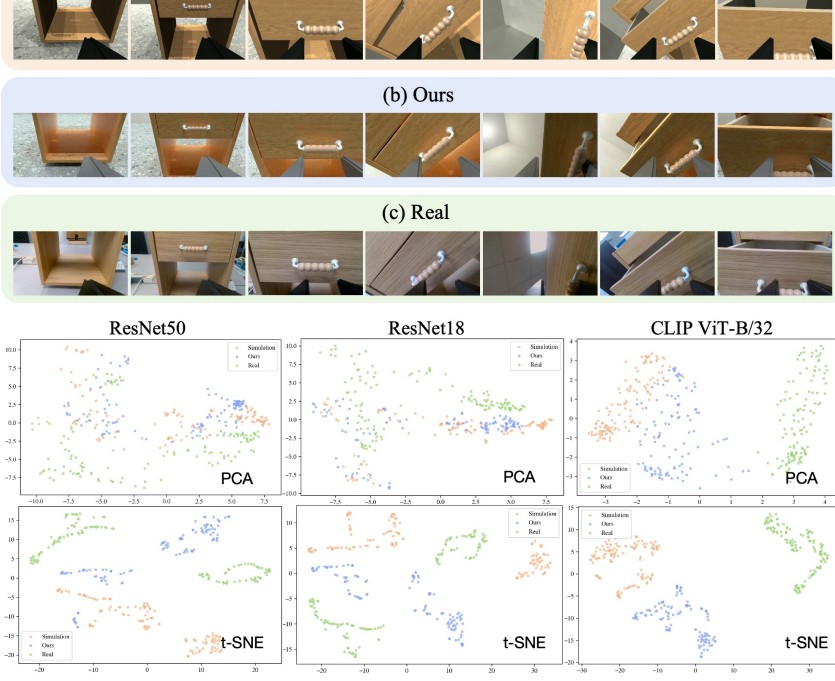

Figure 6: Comparison of simulator renderings, enhanced renderings, and real-world camera frames. PCA and t-SNE projections show that enhanced renderings align more closely with real-world distributions.

| | FID | | | EMD | | |
|---|---|---|---|---|---|---|
| | ResNet-18 | ResNet-50 | ViT-B/32 | ResNet-18 | ResNet-50 | ViT-B/32 |
| Ours | **145.768** | **186.767** | **39.158** | **0.246** | **0.140** | **0.164** |
| Baseline | 146.694 | 193.028 | 47.487 | 0.257 | 0.151 | 0.199 |

Table 2: Comparison of latent distribution distances between simulated and real-world videos. Lower values indicate closer alignment with real data.

## 4 RELIABLE BENCHMARKS AND ANALYSES

Manipulating articulated objects often requires more than simple parallel-jaw grasps. As illustrated in Figure 7, Four characteristic failure modes arise: slip, narrow clearance, mutual interference, and specialized designs. These challenges demand *contact-rich manipulation*, where the end-effector adapts its pose and exploits contact geometry to achieve reliable operation.

Figure 8 shows how Real-IKEA assets reproduce these challenges and enable realistic strategies to overcome them. Thanks to high-fidelity interactive parts and precise collision modeling, our environment supports non-trivial actions such as hooking under a handle or pushing laterally against a knob. By contrast, conventional simulators—whose collision meshes are coarse and often omit holes or curved surfaces—cannot reproduce such strategies. This highlights Real-IKEA as a benchmark that exposes both the difficulty and the potential of contact-rich articulated manipulation.

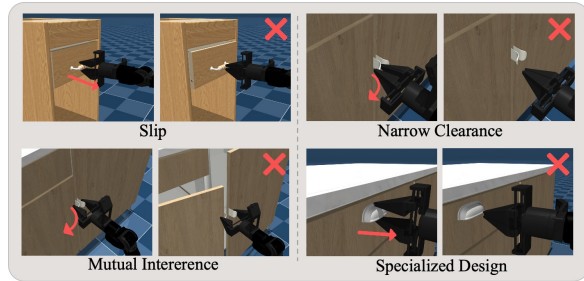

Figure 7: Four characteristic failure modes when robots interact with real articulated objects.

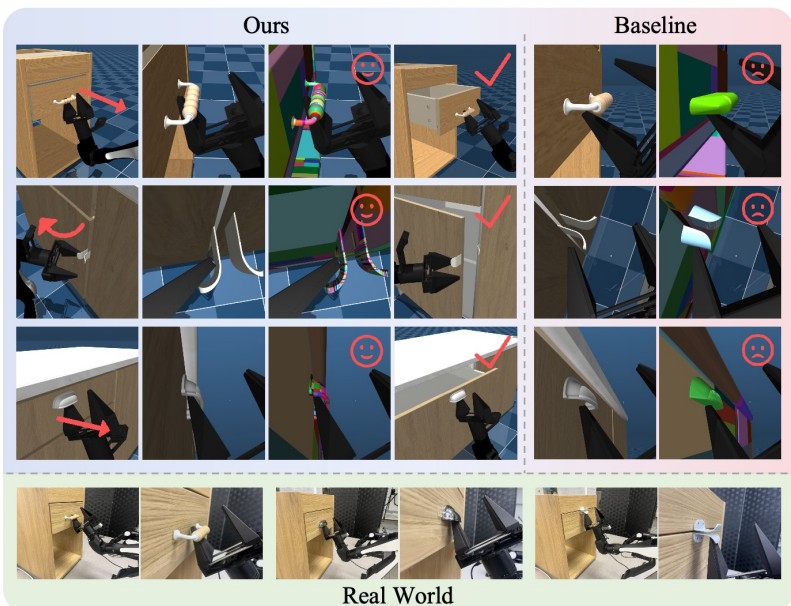

Figure 8: Real-IKEA enables realistic contact-rich strategies consistent with real-world behavior.

### 4.1 CASE STUDY: DRAWER OPENING UNDER JOINT RESISTANCE

We further isolate a canonical task: opening a drawer with a parallel-jaw gripper under varying joint resistance. While seemingly simple, this task becomes challenging once realistic damping and friction are introduced. We configure three resistance modes: *Smooth* (damping = 2, friction = 5), *Normal* (damping = 5, friction = 15), and *High Resistance* (damping = 10, friction = 30).

We compare three representative strategies: (1) **Human teleoperation** in simulation, where an operator completes the task via target-pose teleoperation; (2) **Ground-truth baselines**, in which the gripper is placed directly on the handle or knob (horizontal or vertical grasp) and attempts to pull using friction; and (3) **GraspGen**, a large-scale grasping model that predicts grasp poses conditioned on the ground-truth handle geometry, then attempts to pull. By providing ground-truth perception for all methods, we eliminate sensing errors and isolate their ability to cope with joint resistance.

### 4.2 RESULTS ACROSS HANDLE TYPES

We evaluate three representative handle categories: knobs, finger-pull handles and two-point handles. Results are summarized in Table 3. Success outcomes are categorized as: **F** (Fail: no meaningful progress), **P** (Partial: drawer opens slightly but grasp slips), **S** (Success: drawer fully opens).

**Key findings:** (1) For knobs and finger-pull handles, both ground-truth baselines and GraspGen succeed in smooth settings but performance collapses under normal or high resistance. (2) Teleoperation achieves higher success, yet even humans struggle under high resistance for certain handle and knob

**(a) Knobs**

| Manipulation Policy | Smooth | | | Medium Friction | | | High Friction | | |
|---|---|---|---|---|---|---|---|---|---|
| | F | P | S | F | P | S | F | P | S |
| GraspGen | 0.24 | 0.07 | 0.69 | 0.76 | 0.21 | 0.03 | 1.00 | 0.00 | 0.00 |
| Normal Ways | 0.00 | 0.00 | 1.00 | 0.59 | 0.31 | 0.10 | 1.00 | 0.00 | 0.00 |
| Human Teleoperation | 0.00 | 0.00 | 1.00 | 0.03 | 0.17 | 0.79 | 0.10 | 0.52 | 0.38 |

**(b) Finger-pull handles**

| Manipulation Policy | Smooth | | | Medium Friction | | | High Friction | | |
|---|---|---|---|---|---|---|---|---|---|
| | F | P | S | F | P | S | F | P | S |
| GraspGen | 0.31 | 0.00 | 0.69 | 0.92 | 0.08 | 0.00 | 1.00 | 0.00 | 0.00 |
| Normal Ways | 0.00 | 0.00 | 1.00 | 0.85 | 0.15 | 0.00 | 1.00 | 0.00 | 0.00 |
| Human Teleoperation | 0.00 | 0.00 | 1.00 | 0.00 | 0.04 | 0.96 | 0.00 | 0.65 | 0.35 |

**(c) Two-point handles**

| Manipulation Policy | Smooth | | | Medium Friction | | | High Friction | | |
|---|---|---|---|---|---|---|---|---|---|
| | F | P | S | F | P | S | F | P | S |
| GraspGen | 0.04 | 0.00 | 0.96 | 0.61 | 0.07 | 0.32 | 0.89 | 0.00 | 0.11 |
| Normal Ways | 0.00 | 0.00 | 1.00 | 0.72 | 0.07 | 0.21 | 1.00 | 0.00 | 0.00 |
| Human Teleoperation | 0.00 | 0.00 | 1.00 | 0.00 | 0.00 | 1.00 | 0.00 | 0.00 | 1.00 |

Table 3: Success rate distribution across different manipulation policies under varying resistances.

geometries. (3) For two-point handles, teleoperation achieves perfect success across all resistance levels, including high resistance. Closer inspection shows that teleoperation exploits contact-rich strategies—such as inserting a gripper finger into a handle loop or leveraging curved surfaces—that are not available to conventional baselines.

### 4.3 IMPLICATIONS FOR CONTACT-RICH MANIPULATION POLICIES

Our results yield two core implications for policy design. First, introducing realistic joint resistance and diverse interactive parts sharply increases task difficulty, invalidating friction-dominated strategies that appear effective in simplified simulators. Second, successful behavior hinges on *exploiting contact geometry* to convert actuator effort into opening torque about the hinge axis. In practice, reliable executions favor form-/fixture-closure over pure frictional pulling: inserting a finger into a loop, hooking under a lip, or bracing against curvature increases normal force, improves the moment arm, and aligns the contact wrench with the task goal. Put differently, policies must reason about the moment $\tau = r \times f$ (maximizing the effective arm $r$ and the favorable component of $f$), not just grasp stability.

Human teleoperation shows the task is not solved by a single open-loop pull: operators continuously tweak approach, wrist, and finger placement in response to feedback, especially at contact-state transitions. Slip or stalled motion then cues regrasping/levering, motivating a *closed-loop* formulation that detects these cues, updates pose and contact mode, and exploits geometry to generate hinge torque rather than rely on friction.

## 5 CONCLUSION

We introduced **Real-IKEA**, a dataset and simulation framework that tackles the sim-to-real gap along the three canonical axes—**assets**, **physics**, and **visuals**. Our evaluations demonstrate improved physical fidelity and tighter visual alignment, and show that success under realistic resistance hinges on exploiting geometry to generate hinge torque. It provides a dependable foundation for learning manipulation policies that transfer to contact-rich real-world settings, and a robust benchmark for evaluating contact-rich manipulation.

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

## A  APPENDIX

### A.1  DATASET CONSTRUCTION

Different from object assets collected from existing datasets (Xiang et al., 2020), all the object assets in our dataset are obtained from IKEA. We dedicate significant time and effort to carefully selecting the available object meshes, segmenting them into distinct parts, re-aligning the object mesh coordinate systems, generating collision meshes, and configuring joint parameters.

The construction of a single ready-to-use articulated asset involves the following **six-step processing workflow**:

1. Component Segmentation and Formatting (Visual Mesh Preparation): We manually segment the overall mesh into distinct components (e.g., cabinet base, drawers, doors). These parts are saved as separate meshes. The initial mesh format(GLB) is converted to OBJ format, and corresponding texture maps are generated.

2. Structural Cleaning for Simulation: We manually refine the component meshes by altering the mesh structure and removing features that would interfere with physics simulation. For instance, real-world components like drawer rollers and corresponding tracks must be manually removed or adjusted, as these complex structures often cause problematic self-collision in the simulator.

3. Pre-Collision Scaling and Clipping: Since using Approximate Convex Decomposition(ACD) inevitably causes a slight "swelling" of the mesh, we manually scale or clip specific areas to prevent unreasonable or inter-part collisions after decomposition. Without this step, components like doors or drawers might be "stuck" against the frame (e.g., collision meshes are sealed tightly) and cannot be opened.

4. Collision Mesh Generation: The cleaned meshes are now treated as the final visual meshes. Based on these non-convex meshes, we use the COACD algorithm to generate the corresponding collision meshes. This process generates hundreds or even thousands of sub-meshes (convex primitives), all of which are packaged and managed to serve as the high-fidelity collision geometry for physical contact solving.

5. Assembly and Joint Parameterization: We manually assemble the individual components to form the final articulated object. This crucial step involves accurately determining the joint positions and defining the joint parameters (type, axis, range).

6. Simulation Integration: Finally, the entire object—including the visual meshes, the accurate collision meshes, and all joint information—is assembled and written into a unified XML format that the MuJoCo simulator can correctly load and interpret.

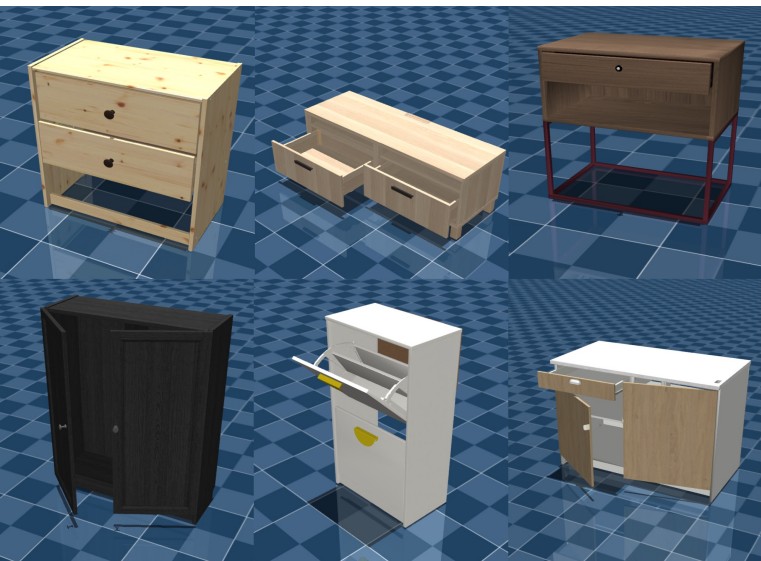

Figure 10: The dataset assets encompass all typical joint types and can be freely combined with interactive components.

### A.2 COLLISION ACCURACY VISUALIZATION

We conducted a visualization of collision accuracy 11. The heatmap results show that the collision models for knobs are highly precise, while for handles, minor errors appear in regions with sharp curvature changes. Overall, the collision models remain sufficiently accurate to support contact-rich manipulation tasks.

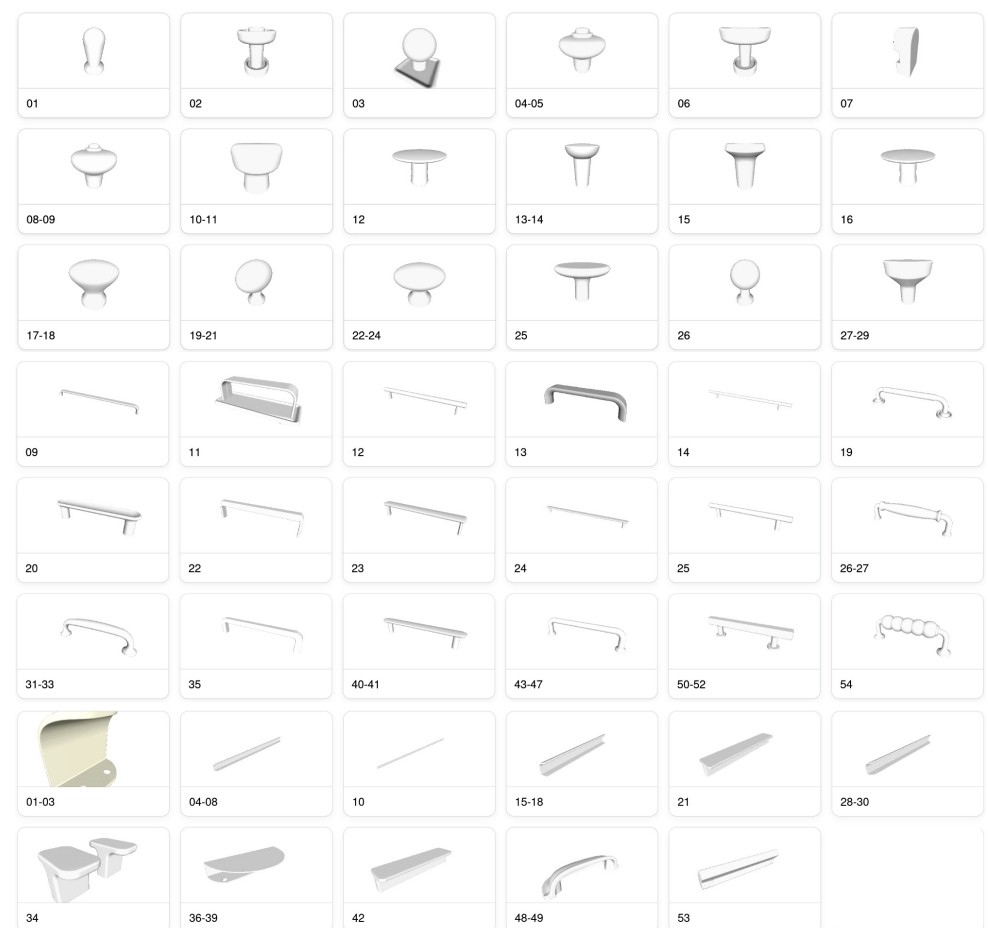

Figure 9: All the shapes of interactive components in Real-IKEA. In our evaluation, we categorize all interactive object parts into three types according to how their geometries affect manipulation strategies. The first type is **knobs** (shown in the first three rows of the figure), the second type is **two-point handles** (rows 4–6), and the third type is **finger-pull handles** (last two rows).

In terms of joint diversity, our dataset further incorporates a subset of IKEA furniture that covers all major joint types and supports flexible composition with interactive parts (Figure 10), thereby facilitating broader evaluation of manipulation policies across diverse articulated structures.

## USE OF LARGE LANGUAGE MODELS

Large Language Models (LLMs) were used solely as writing assistants to help polish grammar and improve clarity of exposition. No content, technical claims, or experimental results were generated by LLMs. All scientific contributions and analyses are the work of the authors.

756
757
758
759
760
761
762
763
764
765
766
767
768
769
770
771
772
773
774
775
776
777
778
779
780
781
782
783
784
785
786
787
788
789
790
791
792
793
794
795
796
797
798
799
800
801
802
803
804
805
806
807
808
809

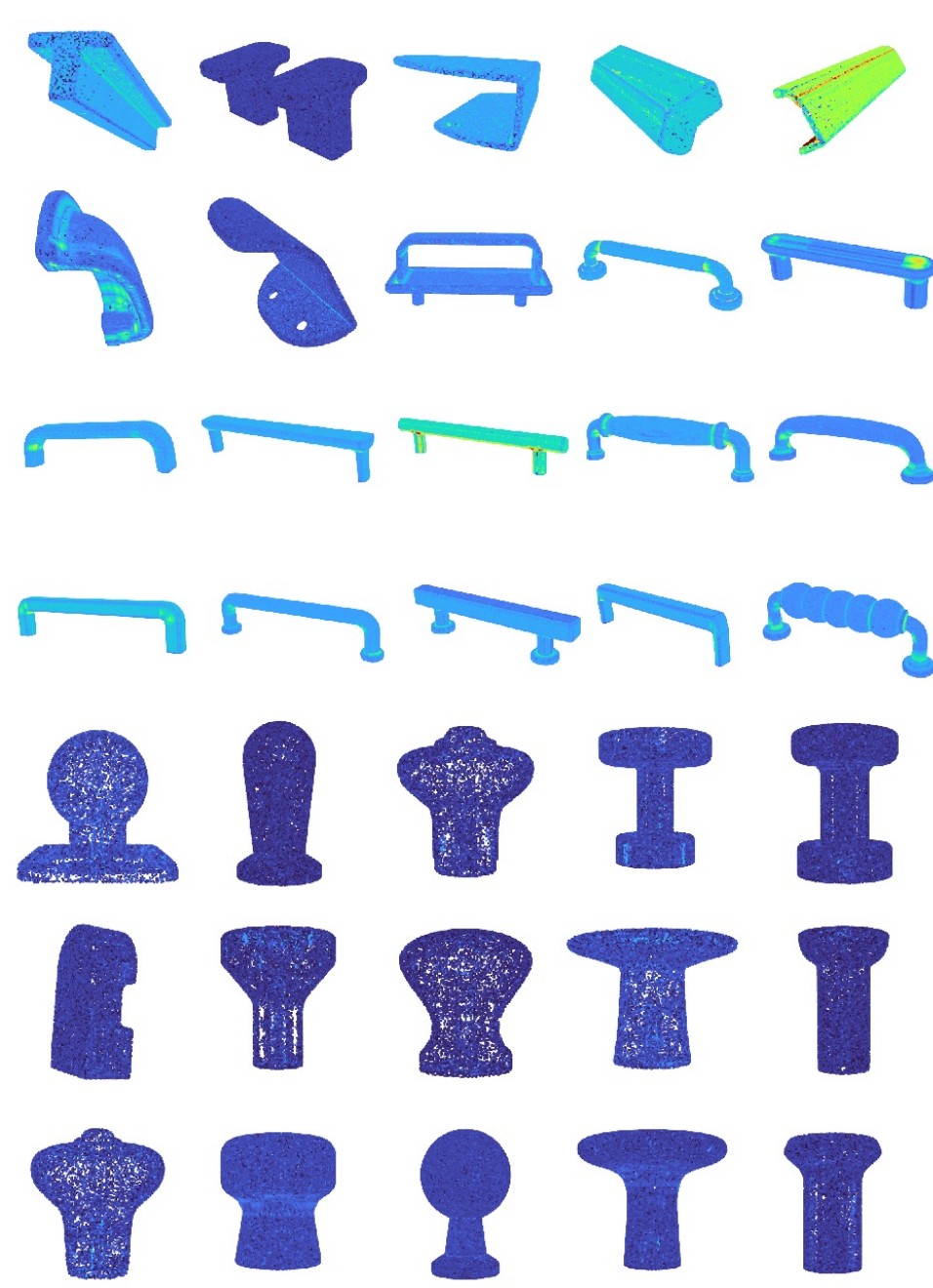

Figure 11: Heatmap visualization of main Real-IKEA interactive components. All of them show $H_{Q \rightarrow P}$