# OpenReview forum: "Real-IKEA : Simulating What Robots Will Really See and Touch"
_ICLR.cc/2026/Conference — Submitted to ICLR 2026_

### Official Review · Reviewer_eEZQ · 2025-10-26

**Soundness:** 3
**Presentation:** 2
**Contribution:** 3
**Rating:** 6
**Confidence:** 4

**Summary:**

The paper introduces Real-IKEA, a benchmark and simulation platform for robot manipulation. Utilizing realistic articulated assets curated from real-world IKEA furniture, employing MuJoCo for physics simulation, and carefully designing the collision meshes via CoACD convex decomposition, Real-IKEA makes a step forward in the field of articulated object manipulation simulation platforms. Authors further accompany this platform with a benchmark contribution to demonstrate the value of Real-IKEA in pushing forward the development of robot articulated object manipulation.

**Strengths:**

- Nice motivation. A realistic and high-performant robot articulated object manipulation platform is a key to advancing the development of robot manipulation policies. This paper identifies three critical challenges in robot manipulation policy sim-to-real transfer and contributes a more realistic simulation platform and benchmark to address the underlying problem.
- Good contributions. The paper makes a solid contribution in aspects of realistic furniture asset curation, accurate convex approximation, and realistic visual rendering.

**Weaknesses:**

- The major weakness is the lack of technical contribution. The authors have made a huge amount of efforts in building the benchmark. However, the technical contribution is weak. To improve the physical simulation fidelity, the paper incorporates two existing techniques/simulators, including CoACD convex approximation and the MuJoCo simulator, with no new decomposition strategies or simulation methods proposed. The solution to improve visual rendering is also to combine existing techniques.
- Restricted coverage of the asset and limited manipulation scenarios. Real-IKEA primarily focuses on drawer-style furniture. The benchmarked manipulation scenarios are mainly pulling out by contacting the handle. The asset is limited in diversity, compared to SAPIEN. Besides, the manipulation scenario is oversimplistic.

**Questions:**

- How about the simulation efficiency when using fine convex decomposed meshes?
- Could it support parallel simulation?
- What does "reality see and touch" mean? Does Real-IKEA reconstruct materials from real-world furniture or does it support tactile simulation as well as tactile policy training?

---

> ### Author Response · Authors · 2025-11-20
>
> # Response to Reviewer eEZQ [1/2]
> We thank the reviewer for the thorough reading and thoughtful feedback. We are particularly grateful that the reviewer recognized the strong motivation of our work in addressing the "underlying problem" of bringing household robotics to reality, as well as the solid contributions in enhancing simulation realism.
>
> We acknowledge the reviewer's concern regarding the lack of technical novelty. While we incorporate powerful existing tools, our core contribution lies in an accuracy-centric design methodology and its systematic application to create an unprecedentedly realistic and challenging benchmark for contact-rich manipulation. Our work's novelty is not in inventing new algorithms for simulation or rendering. Instead, it stems from a **first-principles analysis** of why policies learned in previous simulators fail in the real world.
>
> A critical finding is that **al**l prior articulated object assets only enabled policies to learn **where** to grasp, but crucially, not **how** to grasp. Even when visual meshes featured detailed geometry(grooves, holes) and even without constraints on the robot end-effector pose during learning, the resulting manipulation policies largely converged on simple, friction-dominated parallel grasping motions relative to the target part. This systemic limitation arises because the underlying collision solver only supports convex meshes for contact simulation, forcing non-convex parts to be approximated by their convex hulls during physical computation. Consequently, prior work actually trained end-effectors to interact in the physical simulation with simplistic, solid shapes(like a "cuboid bar" or "semicircular disc").
>
> Our work addresses this necessity head-on:
> - Physical Accuracy: Our case study demonstrates that policies learned on coarse models fail under realistic resistance, while reliable, real-world policies explicitly exploit the geometric features of the object structure for form-closure. This confirms that past manipulation policy was insufficient, and thus, accurate contact modeling for operable parts is a required prerequisite for meaningful research
> - Visual Accuracy: Enhancing the visual pipeline supports the inevitable shift toward vision-based learning methods, which are necessary to perceive and reason about the complex geometric features that enable these advanced manipulation policies.
>
> Starting from this observation, our work provides the necessary pipeline and, for the first time, offers a reliable evaluation foundation for next-generation policies. This constitutes our core novelty and contribution.

---

> > ### Author Response · Authors · 2025-11-20
> >
> > # Response to Reviewer eEZQ [2/2]
> > Below, we address the reviewer’s specific questions.
> > 1. ## How about the simulation efficiency when using fine convex decomposed meshes?
> >    This is a critical question for large-scale data collection.
> >    To quantify the cost of this fidelity, we systematically compared average simulation efficiency (Frames Per Second, or FPS) for various interactive part types, contrasting the efficiency with and without convex decomposition.
> >     | Interactive Part Type | Number of Convex Sub-meshes | Baseline Sim. Efficiency(FPS) | Decomposed Sim. Efficiency(FPS) |
> >     | ------- | ------- | ------- | ------ |
> >     |  Finger-pull handle$_a$  | 24 | 437.88 | 403.83 |
> >     |  Finger-pull handle$_b$  | 224 | 326.13 | 314.29 |
> >     |  Two-point handle$_a$ | 129 | 223.66 | 211.97 |
> >     |  Two-point handle$_b$ | 261 | 309.94 | 299.44 |
> >     |  Knob$_a$ |1258 | 131.64 | 117.41 |
> >     |  Knob$_b$ | 1406 | 148.42 | 145.38 |
> >
> >    The results demonstrate that even with a substantial increase in the geometric complexity of the collision meshes (up to 1406 sub-meshes for certain knobs), the use of fine convex decomposition does not cause a catastrophic drop in simulation efficiency. The FPS remains within a practical range for scalable data collection and policy learning, validating our design choice.
> >
> > 2. ## Could it support parallel simulation?
> >    Yes, Real-IKEA fully supports parallel simulation.
> >    - Physics Simulation: All our assets are integrated into standard XML formats, which can be solved using MuJoCo MJX physics solver. The mjx engine is explicitly designed for GPU-based parallel simulation, making policy training and large-scale data generation highly efficient.
> >    - Rendering Pipeline: Our hybrid rendering pipeline records all environment state changes during real-time data collection. The subsequent offline high-quality re-rendering is entirely based on these logged state changes, meaning it can be executed efficiently in a GPU parallel fashion on a render farm.
> > 3. ## What does "reality see and touch" mean? Does Real-IKEA reconstruct materials from real-world furniture or does it support tactile simulation as well as tactile policy training?
> >    The phrase "really touch"(as featured in the paper's title) refers to physical interaction fidelity(asset and physics gaps). This addresses the core weakness of prior datasets where simplified collision meshes fail to support fundamental contact-rich policies. We emphasize that our framework is the first articulated object asset collection that supports policies like hooking, leveraging, and bracing by accurately preserving the fine-grained geometry of holes and grooves. Our case study showed that policies relying on simple friction-dominated parallel grasping fail under realistic joint resistance. This is the essence of "really touching" the object—interacting with its true geometry to generate torque ($\tau=r\times f$) rather than just friction.
> >
> >    Regarding your specific query on material properties:
> >    - Material and Surface Details: We currently do not support realistic soft body deformation, material hardness, or fine surface roughness details beyond the geometric mesh and calibrated joint resistance/friction. While these details may eventually aid in policy learning, we currently believe that the missing accurate contact geometry and joint dynamics are the most critical omissions preventing the learning of real-world manipulation methods.
> >    - Tactile/Contact Simulation: The underlying MuJoCo physics engine naturally supports collision-based tactile simulation. While our current benchmark focuses on the manipulation task success, the detailed contact force/torque information is available, and the framework can readily be used for training tactile-based policies that leverage these precise collision models.
> >
> >     In conclusion, we believe that Real-IKEA makes a unique and essential contribution by prioritizing the realism of interactable geometry and dynamics (joint resistance) at scale, which is fundamentally necessary for the learning of manipulation policy which can be used in real world.

---

### Official Review · Reviewer_miW6 · 2025-10-31

**Soundness:** 3
**Presentation:** 3
**Contribution:** 2
**Rating:** 2
**Confidence:** 4

**Summary:**

This paper introduces a new dataset of 3D assets of IKEA cabinets with realistic visuals and collision geometry, diverse coverage, and configurable joint resistance.

**Strengths:**

This dataset is useful and could help the community's research on contact-rich manipulation of articulated objects based on simulation. The more accurate collision geometry makes simulation more accurate and shrinks the sim-to-real gap.

**Weaknesses:**

The contribution of this paper is majorly the new dataset. It definitely has value, as I said, but there are few innovations or insights either to learning in general or robot manipulation. The construction of the more accurate collision geometry and the higher-fidelity visualization are using off-the-shelf methods. The physics parameters (the resistance) are manually specified, which is not grounded in the real data. This paper might be a better fit for robotics venues.

**Questions:**

Is the bidirectional surface-deviation metric just chamfer distance?

I didn't quite understand Figure 6. How did you get the corresponding images in the real-world? The teleoperation is done in simulation, right?

A lot of details of the case study is missing. How is success defined? Are multiple attempts allowed? Does human teleoperator know the resistance level of the task in prior? How forceful is the grasp, or it is not in the parameter space of grasps?

**Details Of Ethics Concerns:**

I'm not sure if there would be copyright concerns for IKEA furniture.

---

> ### Author Response · Authors · 2025-11-20
>
> # Response to Reviewer miW6 [1/3]
> We thank the reviewer for the time spent on the review and for acknowledging the value of our contribution in creating a useful dataset that "makes simulation more accurate and shrinks the sim-to-real gap."
>
> We wish to address the concerns raised in the weaknesses section regarding technical innovation/insights and the grounding of physics parameters.
> ## Weakness 1: Few innovations or insights to learning in general or robot manipulation.
> We **strongly disagree** with the assertion that our work provides "few innovations or insights." Our paper is built upon a **fundamental insight** validated by our case study: the prevailing parallel grasping policies learned in existing simulations are fundamentally unreliable in the real world when realistic joint resistance is introduced.
> - **Insight**: We provide a rigorous explanation for this failure, tracing it back to a **critical incompatibility between past asset design and the physics solver's requirements**. Due to the physics engine's reliance on convex collision meshes, all prior articulated object datasets forced agents to learn interactions with grossly simplified, convex approximations (like a "cuboid bar" or "semicircular disc"), regardless of the visual mesh's complexity. Consequently, they learned where to grasp (position) but lacked the ability to learn how to grasp(better contact pose) that exploits geometric features for leverage.
> - **Innovation**: Our innovation lies not in creating a new algorithm for simulation or rendering, but in applying first-principles reasoning to solve a critical, foundational gap in robotics research. We are the **first** to:
>   1. Diagnose this contact-geometry failure mode systematically.
>   2. Develop and execute the complete, high-effort pipeline—from digital sourcing of real products to high-fidelity decomposition and physics calibration—necessary to restore accurate contact modeling for non-convex interactive parts.
>   3. Empirically demonstrate via teleoperation that this increased physical fidelity unlocks the learning of effective, form-closure-based policies (e.g., hooking under a lip or bracing against a curve) required to overcome realistic joint resistance.
>
> Our work establishes the necessary infrastructure and **first reliable evaluation platform** for policies that aim to leverage geometry for torque, pushing the community beyond friction-only methods. This is a **foundational, non-trivial, and highly innovative** contribution.
>
> ## Weakness 2: The physics parameters (the resistance) are manually specified, which is not grounded in the real data.
> **We have never stated that the joint-resistance parameter is manually specified**. As explained in Table 1 and Line 219, it is a **“configurable joint resistance.”** In the case study, we configure three resistance modes to compare how the manipulation policy performs under different levels of resistance. We respectfully **disagree** with treating this “configurable joint resistance” as a weakness; on the contrary, it is an intentional and essential design feature that enables robust policy learning.
>
> While measuring and fixing a specific resistance parameter from a real-world object (a real2sim approach) seems "grounded," it severely limits the utility for research. Our ultimate goal is not to train a robot to open a single cabinet with a fixed resistance, but to **learn robust policies that can handle thousands of cabinets with varying, unknown resistance levels**.
>
> This necessity directly motivates our design choice:
>
> - **Domain Randomization** : Policy learning, especially via Reinforcement Learning, requires Domain Randomization over critical parameters like joint resistance. Our configurable resistance allows researchers to sample continuously from a broad range of resistance parameters, which is essential for training policies robust to real-world friction variations.
>
> - **Real-World Variability and Gravity Dependence**: We further note that in the real world, joint resistance is not a static property. For drawer slides, resistance mechanisms (like slope in the roller track or simple sliding friction) often depend on the total **mass** of the drawer. Since the mass is highly variable based on the items placed inside, fixing a single, seemingly accurate resistance value is ultimately non-generalizable and meaningless. The resistance level required to open the cabinet is not constant in real life.
>
> By providing an interface to precisely specify and vary joint resistance, we offer the optimal method for researchers to employ Domain Randomization, thus contributing to the learning of truly robust manipulation policies.

---

> ### Author Response · Authors · 2025-11-20
>
> # Response to Reviewer miW6 [2/3]
> We appreciate the reviewer's specific questions, which allow us to clarify key technical and experimental details of Real-IKEA.
> ## Question 1: Is the bidirectional surface-deviation metric just Chamfer Distance?
> The Chamfer Distance(CD) is a well-defined metric for comparing the shape similarity between two point clouds, where both point clouds are treated as equally important. Our **newly defined** distance calculation process is similar to the Chamfer Distance, but we divide it into two directional components. Furthermore, we intentionally chose this approach and utilized the squared Euclidean distance in the average calculation, rather than directly using the standard Chamfer Distance, because in our area of focus —— “whether the equivalent collision mesh is sufficiently close to the visual mesh in the simulator”—— the two point clouds are not of equal status: the **visual mesh ($P$) is the standard**.
>
> To evaluate the specific impact of shape approximation on contact simulation, simply looking at the single CD value is far from sufficient. This number cannot tell you where the structure deviates significantly, or if this deviation affects manipulation policies in the simulation. Therefore, we used the **point-wise results** of the directionally separated distance to plot detailed heatmaps (Figure 5 ) to show which specific parts of the operable components exhibit deviations, and how this impacts contact simulation. This analysis also reveals that directly using the original meshes results in severe errors that critically affect manipulation performance, a detail that cannot be discerned from a single distance value, thus demonstrating the necessity of our collision mesh processing.
>
> ## Question 2: I didn't quite understand Figure 6. How did you get the corresponding images in the real-world? The teleoperation is done in simulation, right?
> The teleoperation was performed in the simulated virtual environment. It is because the teleoperation was conducted virtually that we were able to record the precise pose of the robot arm for every frame of the trajectory. Since we also know the fixed environmental parameters in the simulation, such as the relative position of the robot arm and the cabinet, we were able to set up and fix the robot arm and the cabinet in the real world at the identical relative configuration. Then, we replayed the recorded teleoperation trajectory in real world and obtain the corresponding images.
>
> ## Question 3: A lot of details of the case study is missing. How is success defined? Are multiple attempts allowed? Does human teleoperator know the resistance level of the task in prior? How forceful is the grasp, or it is not in the parameter space of grasps?
> - How is success defined?
>     As written in the Line428 of original paper, "Success outcomes are categorized as: F (Fail: no meaningful progress), P (Partial: drawer opens slightly but grasp slips), S (Success: drawer fully opens)." We did not simply categorize results as success or fail; we defined three distinct outcomes:
>     - "No meaningful progress" (F ): The drawer opens insufficiently to place any item inside.
>     - "Drawer opens slightly but grasp slips" (P ): The drawer opens to 1/4 of its maximum extent, and the grasp slips afterwards.
>     - "Drawer fully opens" (S ): The drawer reaches the maximum extent allowed by the joint limits.
>
> - Are multiple attempts allowed?
>     Only one pull was allowed. However, the human teleoperator could continuously adjust the grasping position before initiating the pull.
>
> - Does human teleoperator know the resistance level of the task in prior?
>     The teleoperator did not know the resistance level beforehand but was instructed to use the manipulation pose they deemed most reliable.
>
> - How forceful is the grasp, or it is not in the parameter space of grasps?
>     The magnitude of the grasping force is not a fixed value, as the real robot's low-level control utilizes PD control. The final grasping force in our experiment is determined by a fixed set of PD control parameters. This is position-based control. We used the exact same set of PD control parameters throughout the entire experiment for all comparisons to ensure fairness.

---

### Official Review · Reviewer_e7P3 · 2025-10-31

**Soundness:** 2
**Presentation:** 2
**Contribution:** 2
**Rating:** 2
**Confidence:** 3

**Summary:**

This paper introduces Real-IKEA, an articulated object dataset for contact-rich articulated object manipulation. The assets are constructed from real IKEA furniture bases paired with 83 authentic IKEA handles and knobs, yielding 1,079 articulated configurations. The work emphasizes the three axis of sim2real gap: object geometry, physics realism, and visual fidelity, by COACD-based collision modeling and a hybrid rendering pipeline. The paper provides quantitative evaluations of collision modeling accuracy, visual realism, and manipulation success rates under varying joint resistance.

**Strengths:**

1. The use of real IKEA furniture bases and accurately modeled handles/knobs gives the dataset strong grounding in real-world distributions.
2. The bidirectional metric and use of the COACD algorithm for convex decomposition improves the collision fidelity for handles and knobs, which are shown to be important for contact-rich manipulation.
3. The two-stage rendering process (real-time + offline re-rendering) narrows the vision gap, supported by quantitative FID/EMD improvements and t-SNE visualizations.
4. The human-in-the-loop evaluation is interesting. The inclusion of teleoperation studies and human performance benchmarks provides insight into how accurate handle modeling influences manipulation strategies.
5. The case study on drawer opening under varying resistance levels is an interesting design that highlights the limitations of friction-dominated policies and the importance of accurate collision modeling.

**Weaknesses:**

1. Limited Downstream Validation: Although the dataset shows improved FID/EMD and collision accuracy, it remains unclear how much these improvements translate to real policy performance, e.g. visual sim2real manipulation policy transfer. Including at least one downstream comparison (training in Real-IKEA and testing on a real robot) would strengthen the claim.
2. Comparison Scope: While comparisons with PartNet-Mobility and AdaManip are provided, a direct quantitative or qualitative comparison with GAPartManip (ICRA 2024) or similar large-scale articulated manipulation datasets would help contextualize the novelty.
3. Missing experimental details: The experimental details for Tables 2 and 3 are incomplete. In particular, sample sizes and human study protocols are missing.
4. Missing related works: Since one major contribution of the paper is to mitigate the sim2real gap, it should include the Real2Sim2Real line of work in robotics that aim to mitigate the visual[1,2] and/or physics[3] gaps. Distinguish Real-IKEA to this body of work would clarify its contribution.

[1] Li, Xinhai, et al. "Robogsim: A real2sim2real robotic gaussian splatting simulator." arXiv preprint arXiv:2411.11839 (2024).

[2] Yu, Justin, et al. "Real2render2real: Scaling robot data without dynamics simulation or robot hardware." arXiv preprint arXiv:2505.09601 (2025).

[3] Lim, Vincent, et al. "Real2sim2real: Self-supervised learning of physical single-step dynamic actions for planar robot casting." 2022 International Conference on Robotics and Automation (ICRA). IEEE, 2022.

**Questions:**

1. For Table 2 (FID/EMD): How many real-world videos were used for computing the distribution distances? Were these captured in a controlled lighting environment, or across multiple real settings?
2. For Table 3 (Success Rates): How many objects, human subjects, and trials per condition were used to compute the success-rate distributions?
3. How were the damping and friction parameters (2/5, 5/15, 10/30) determined. Is it through physical measurement on real furniture, or empirical tuning in simulation?
4. The paper mentioned the knobs and handles are systematically combined with the base cabinets, but there are missing details on how the attach locations and rotation axises are generated.

---

> ### Author Response · Authors · 2025-11-21
>
> # Response to Reviewer e7P3 [1/3]
> We thank the reviewer for careful assessment and generous recognition of the strong grounding of our dataset , the importance of our collision fidelity , the effectiveness of our hybrid rendering pipeline , and the valuable insight provided by case study.
>
> We address the weaknesses below, clarifying the scope of our contribution.
> ## Weekness 1:
> We acknowledge that including a fully trained policy transferred to a real robot would provide the most direct validation. However, achieving this is currently non-trivial due to the specific, high-level goal of our platform: to enable policies that can learn rubust contact-rich strategies based on handle geometry.
> - Policy Gap: Current state-of-the-art algorithms are still insufficient for this task. Using state-based Reinforcement Learning (RL) is difficult because handle geometry cannot be easily abstracted into a state vector. Vision-based RL remains challenging to train robustly. Affordance-based or Imitation Learning (IL) methods often rely on predefined, fixed grasping poses, which fundamentally contradicts our goal of enabling the agent to learn to adjust its pose based on diverse, specific handle geometries.
> - Goal of Real-IKEA: Our primary contribution is providing the essential, missing physics-accurate platform necessary for such an algorithm to exist and succeed. We confirm that we are actively working towards finding an effective policy structure, and we believe Real-IKEA is the best platform available for this significant research breakthrough. Demonstrating this breakthrough policy transfer is currently infeasible, but we have provided the foundational tool needed for the community to pursue it.
>
> ## Weekness 2:
> We appreciate the reviewer's suggestion to include comparisons with GAPartManip. However, we believe there may be a **minor oversight of reviewer** regarding the asset source:
> - The GAPartManip paper(ICRA 2025) explicitly states that its assets are inherited from the GAPartNet dataset ("Our GAPartManip inherits 918 object instances across 19 categories from the previous **GAPartNet dataset**").
> - The GAPartNet paper states that its assets are gathered from PartNet-Mobility and AKB-48 ("we present GAPartNet, a large-scale interactive part-centric dataset where we gather 1,166 articulated objects from the **PartNet-Mobility dataset** and the **AKB-48 dataset**").
> Since the AKB-48 dataset remains unavailable for download, and our careful inspection reveals GAPartNet assets purely relies on PartNet-Mobility dataset, our quantitative and qualitative comparisons with PartNet-Mobility and AdaManip are sufficient to contextualize our novelty. Critically, all assets in Real-IKEA are new and were not available in any of prior works, **representing our unique contribution** to the community.
>
> ## Weekness 3:
> We acknowledge the missing experimental details. We will provide a comprehensive breakdown of the sample sizes and protocols in the response to the questions below.
>
> ## Weekness 4:
> We appreciate the suggestion as it helps define our unique contribution in the context of mitigating the sim2real gap.
> - Visual Reconstruction: Robogsim and Real2render2real focus on visual fidelity, using 3D Gaussian Splatting for scene reconstruction. Their visual reality comes from 3D Gaussian Splatting. However, this Gaussian Splatting approach is impractical for scaling to the thousands of different object combinations required for robust articulated object manipulation task.
>     Furthermore, the Real2render2real authors give up combining with physics engines and note that "In early experiments, we explored physics engines for real-to-sim-to-real data generation but found that with imperfect or unrefined real-to-sim assets, simulated dynamics often diverged from real-world behavior—especially in gripper-object interactions, where issues like interpenetration and unrealistic collisions were common.". This observation strongly reinforces our contribution: we invested immense effort into refining the Real2Sim physical assets, making our assets compatible with accurate physics engines(MuJoCo)—a step these methods bypassed.
> - Physics : This work primarily focuses on calibrating robot and environmental dynamics parameters (e.g., friction, robot dynamics). Our work, however, focuses on ensuring the physical accuracy of the object itself—specifically, the high-fidelity contact geometry and object-specific joint resistance—which is a prerequisite for any downstream physics calibration. Our innovation ensures the object interaction is fundamentally correct before policy training begins.

---

> ### Author Response · Authors · 2025-11-22
>
> # Response to Reviewer e7P3 [2/3]
> We appreciate the reviewer's detailed questions and rigorous examination of our experimental protocol.
>
> ## Question 1: For Table 2 (FID/EMD): How many real-world videos were used for computing the distribution distances? Were these captured in a controlled lighting environment, or across multiple real settings?
> The computation of distribution distances relied on three long video sequences(over ten minutes each), ensuring sufficient views from different directions . We ensured that the robot poses and object positions in all three videos were frame-by-frame aligned.
>
> Regarding the light and background environment, we acknowledge the difficulty of precisely replicating complex real-world lighting in simulation. Therefore, we performed a new experiment where we standardized the ambient lighting and background color (to the same wall and floor) in both the raw simulator and our offline renderer, without altering the real-world lighting. This ensures a fair comparison focused on mesh and shading fidelity. The result is shown in Figure 6 and Table 2 of new PDF. These results show that while the improvement under ResNet encoders is slightly reduced compared to the original setup, the enhancement observed under the ViT-B/32 encoder is more pronounced.
>
> ## Question 2: For Table 3 (Success Rates): How many objects, human subjects, and trials per condition were used to compute the success-rate distributions?
> The success rates depend significantly on the manipulation strategy used for each interactive part geometry. Therefore, we selected 10 human teleoperators to participate in this experiment. Each operator was tasked with operating all 83 unique interactive parts(Knobs, Finger-pull handles, and Two-point handles) under all resistance conditions, but only had one pulling opportunity per part. The teleoperator did not know the resistance level beforehand but was instructed to use the manipulation pose they deemed most reliable.
>
> ## Question 3: How were the damping and friction parameters (2/5, 5/15, 10/30) determined. Is it through physical measurement on real furniture, or empirical tuning in simulation?
> Our assets have configurable joint resistance, which allows the resistance parameters to be randomly initialized when loading the object during Reinforcement Learning training (Domain Randomization).
>
> In the case study, we empirically configured three resistance modes (Smooth, Medium Friction, High Friction) to compare how the manipulation policy performs under different levels of resistance. This approach was taken intentionally after careful consideration: We note that in the real world, joint resistance is not a static property. For drawer slides, resistance mechanisms often depend on the total mass of the drawer. Since the mass is highly variable based on the items placed inside, fixing a single, seemingly accurate resistance value is ultimately non-generalizable and meaningless. The resistance level required to open the cabinet is not constant in real life.

---

> ### Author Response · Authors · 2025-11-22
>
> # Response to Reviewer e7P3 [3/3]
> ## Question 4: The paper mentioned the knobs and handles are systematically combined with the base cabinets, but there are missing details on how the attach locations and rotation axises are generated.
> The construction of a single ready-to-use articulated asset involves the following **six-step processing workflow** (We will supplement the appendix with these detailed steps):
> 1. **Component Segmentation and Formatting (Visual Mesh Preparation)**: We manually segment the overall mesh into distinct components (e.g., cabinet base, drawers, doors). These parts are saved as separate meshes. The initial mesh format(GLB) is converted to OBJ format, and corresponding texture maps are generated.
> 2. **Structural Cleaning for Simulation**: We manually refine the component meshes by altering the mesh structure and removing features that would interfere with physics simulation. For instance, real-world components like drawer rollers and corresponding tracks must be manually removed or adjusted, as these complex structures often cause problematic **self-collision** in the simulator.
> 3. **Pre-Collision Scaling and Clipping**: Since using Approximate Convex Decomposition(ACD) inevitably causes a slight "swelling" of the mesh, we manually scale or clip specific areas to prevent unreasonable or inter-part collisions after decomposition. Without this step, components like doors or drawers might be **"stuck"** against the frame (e.g., collision meshes are sealed tightly) and cannot be opened.
> 4. **Collision Mesh Generation**: The cleaned meshes are now treated as the final visual meshes. Based on these non-convex meshes, we use the COACD algorithm to **generate the corresponding collision meshes**. This process generates hundreds or even thousands of sub-meshes (convex primitives), all of which are packaged and managed to serve as the high-fidelity collision geometry for physical contact solving.
> 5. **Assembly and Joint Parameterization**: We manually assemble the individual components to form the final articulated object. This crucial step involves accurately determining the joint positions and defining the joint parameters (type, axis, range).
> 6. **Simulation Integration**: Finally, the entire object—including the visual meshes, the accurate collision meshes, and all joint information—is assembled and written into a unified XML format that the MuJoCo simulator can correctly load and interpret.
>
> We assemble the base cabinets with the interactive parts(Knobs, Finger-pull handles, and Two-point handles) in the last two steps. We manually calibrated the attachment locations to be visually and physically coherent, often based on standard real-world assembly patterns. For handles and knobs, the attachment locations are consistently placed on the drawer/door surface. If a user wishes to substitute a handle, the attachment location and the joint parameters remain constant; only the mesh index within the formatted XML description needs to be replaced, which can be easily automated using scripting.

---

### Official Review · Reviewer_WRPG · 2025-11-01

**Soundness:** 2
**Presentation:** 2
**Contribution:** 2
**Rating:** 2
**Confidence:** 3

**Summary:**

This paper proposes Real-IKEA, a dataset and simulation framework based on real IKEA furniture, which mitigates the asset gap, physics gap and vision gap in simulation environments. For asset generation, the authors combine reusable handles with base cabinet units to create the Real-IKEA dataset. Building on this dataset, the framework also enables high fidelity physical interaction simulation and high-quality visual rendering by reconstructing collision meshes with the COACD algorithm and using a hybrid rendering pipeline. In addition, the authors test different manipulation policies in their simulation environment for validation.

**Strengths:**

1. The proposed Real-IKEA dataset is valuable in robotics research as its assets closely match real-world asset distributions. This helps mitigate the sim-to-real gap in robotics policy deployment.
2. Multiple quantitative evaluations of asset quality and physical-interaction fidelity are helpful for understanding the characteristics of the proposed framework.

**Weaknesses:**

1. The paper does not present many novel contributions. From Section A.1, the object meshes were already available before this work, and the dataset creation is mostly mesh selection and segmentation. Also, for the claimed contributions in realistic physical interaction and visual rendering, the benefits come from using prior methods such as the COACD algorithm and combining existing renderers, which are not contributions of this paper itself.
2. The evaluation in Figure 6 is not fair, as the background colors in the baseline simulator and the real-world frames differ a lot, while the background color of the proposed framework and the real-world frames is similar, which would influence the FID evaluation. Since background color can be easily set in simulators, a fairer comparison is to set the background color of all simulators the same and report results in that setting.
3. There is no citation of the COACD algorithm on Line 254.

**Questions:**

1. How do the proposed framework’s physical interaction realism and visual fidelity compare to prior simulation environments? Are there quantitative results?
2. Are the original meshes for the dataset publicly available and what is the detailed procedure of creating the dataset? These aspects should also be discussed in the paper.

---

> ### Author Response · Authors · 2025-11-21
>
> # Response to Reviewer WRPG [1/2]
> We thank the reviewer for the thorough reading of our paper and for recognizing the value of the Real-IKEA dataset in mitigating the sim-to-real gap, as well as the utility of our quantitative evaluations.
>
> We wish to address the concerns raised in the weaknesses section.
> ## Weekness 1:
> - We believe there is a **misunderstanding regarding the availability of our assets**. We must clarify that the core object meshes for our asset collection were **NOT** publicly available or present in any prior open-source work before our project. As stated in **Section A.1,**"**Apart from** object assets collected from existing datasets..."(We will use "different from" to avoid misunderstanding in new PDF). Our contribution is not merely selection and segmentation of existing assets, but rather the creation of a new, high-fidelity digital twin library sourced from real IKEA products and subjected to a complex processing pipeline(segmentation, coordinate alignment, precise collision mesh generation, and joint configuration).
> - While we incorporate powerful existing tools, our core contribution lies in an accuracy-centric design methodology and its systematic application to create an unprecedentedly realistic and challenging benchmark for contact-rich manipulation. Our work's novelty is not in inventing new algorithms for simulation or rendering. Instead, it stems from a **first-principles analysis** of why policies learned in previous simulators fail in the real world.
>
>     A critical finding is that **al**l prior articulated object assets only enabled policies to learn **where** to grasp, but crucially, not **how** to grasp. Even when visual meshes featured detailed geometry(grooves, holes) and even without constraints on the robot end-effector pose during learning, the resulting manipulation policies largely converged on simple, friction-dominated parallel grasping motions relative to the target part. This systemic limitation arises because the underlying collision solver only supports convex meshes for contact simulation, forcing non-convex parts to be approximated by their convex hulls during physical computation. Consequently, prior work actually trained end-effectors to interact in the physical simulation with simplistic, solid shapes(like a "cuboid bar" or "semicircular disc").
>
>     Our work addresses this necessity head-on:
>     - Physical Accuracy: Our case study demonstrates that policies learned on coarse models fail under realistic resistance, while reliable, real-world policies explicitly exploit the geometric features of the object structure for form-closure. This confirms that past manipulation policy was insufficient, and thus, accurate contact modeling for operable parts is a required prerequisite for meaningful research
>     - Visual Accuracy: Enhancing the visual pipeline supports the inevitable shift toward vision-based learning methods, which are necessary to perceive and reason about the complex geometric features that enable these advanced manipulation policies.
>
>     Starting from this observation, our work provides the necessary pipeline and, for the first time, offers a reliable evaluation foundation for next-generation policies. This constitutes our core novelty and contribution.
> ## Weekness 2:
> We sincerely thank the reviewer for pointing out the potential influence of background color on the FID evaluation in Figure 6. This is a crucial observation regarding experimental fairness.
>
> We have re-run the experiments with the background color of all simulators standardized to a uniform white wall and the same floor to ensure a fair comparison based solely on object realism. The new results are presented in Figure 6 and Table 2 of new PDF.
>
> The new results clearly demonstrate that even with the background standardized across all simulators, our hybrid rendering pipeline consistently achieves lower FID and EMD scores than the baseline simulator. This confirms that our approach narrows the visual domain gap.
> ## Weekness 3:
> We confirm that the citation for the COACD algorithm is provided at its first mention in Line 173. Subsequent mentions of a method often omit the citation for reading clarity, which aligns with standard academic practice. However, we deeply appreciate the reviewer's attention to detail and will ensure that the citation is added on Line 254 in the final version to fully address this suggestion.

---

> ### Author Response · Authors · 2025-11-26
>
> # Response to Reviewer WRPG [2/2]
> We appreciate the reviewer's specific questions, which allow us to clarify details.
> ## Question 1:
> Yes, we provide extensive quantitative results in Sections 3.2 and 3.3 comparing Real-IKEA against prior assets and environments:
> - **Physical Interaction Realism**:
>     Compared to prior work, the enhancement in our physical contact accuracy is significant and visible. Past assets, even if they visually appear to have holes and grooves, were computed using convex hulls for collision, resulting in meshes with large geometric differences(as the rightmost column in Figure 5). Our primary advance is in addressing this low physical accuracy.
>
>     We introduced a bidirectional surface-deviation metric to quantitatively compare the geometric accuracy of our collision meshes against baseline collision meshes. Figures 4,5,11 clearly show that our modeling substantially reduces the average deviation (distance is generally an order of magnitude lower) and preserves critical non-convex features essential for contact. Figures 5 and 11 use heatmaps to display the error, which is the most intuitive way to show the point-wise accuracy.
> - **Visual Fidelity**:
>     We quantify the visual alignment between simulated and real-world camera frames using Fréchet Inception Distance (FID) and Earth Mover's Distance (EMD) across multiple encoders. Table 2 shows that our hybrid rendering pipeline substantially reduces both FID and EMD compared to raw simulator outputs. Lower values of $D_{FID}$ and $D_{EMD}$ indicate a smaller visual gap and closer alignment with real data.
> ## Question 2:
> These initial meshes were in a very raw state (e.g., they lacked joint definitions, component segmentation, and were not in a format compatible with physics simulators), rendering them far from usable for robotic interaction. Therefore, we custom-built a comprehensive mesh processing pipeline.
>
> The construction of a single ready-to-use articulated asset involves the following **six-step processing workflow** (We will supplement the appendix with these detailed steps):
> 1. **Component Segmentation and Formatting (Visual Mesh Preparation)**: We manually segment the overall mesh into distinct components (e.g., cabinet base, drawers, doors). These parts are saved as separate meshes. The initial mesh format(GLB) is converted to OBJ format, and corresponding texture maps are generated.
> 2. **Structural Cleaning for Simulation**: We manually refine the component meshes by altering the mesh structure and removing features that would interfere with physics simulation. For instance, real-world components like drawer rollers and corresponding tracks must be manually removed or adjusted, as these complex structures often cause problematic **self-collision** in the simulator.
> 3. **Pre-Collision Scaling and Clipping**: Since using Approximate Convex Decomposition(ACD) inevitably causes a slight "swelling" of the mesh, we manually scale or clip specific areas to prevent unreasonable or inter-part collisions after decomposition. Without this step, components like doors or drawers might be **"stuck"** against the frame and cannot be opened.
> 4. **Collision Mesh Generation**: The cleaned meshes are now treated as the final visual meshes. Based on these non-convex meshes, we use the COACD algorithm to **generate the corresponding collision meshes**. This process generates hundreds or even thousands of sub-meshes (convex primitives), all of which are packaged and managed to serve as the high-fidelity collision geometry for physical contact solving.
> 5. **Assembly and Joint Parameterization**: We manually assemble the individual components to form the final articulated object. This crucial step involves accurately determining the joint positions and defining the joint parameters (type, axis, range).
> 6. **Simulation Integration**: Finally, the entire object—including the visual meshes, the accurate collision meshes, and all joint information—is assembled and written into a unified XML format that the MuJoCo simulator can correctly load and interpret.
>
> Key Distinguishing Feature (**Novelty**)
> A critical point to note is that all previous articulated object assets relied on only one set of object meshes. This single mesh set served both for rendering (as the visual mesh) and for calculating physical interactions (as the collision mesh). While the visual rendering was fine, using the same mesh for collision led to the high physical inaccuracy we discussed—the appearance of the object in the simulator was drastically different from the geometry used to resolve physical interaction.
>
> Real-IKEA is the **first** asset collection to generate two distinct, corresponding sets of meshes: one for rendering (visual mesh) and a separate, highly accurate set for physical resolution (collision mesh). We believe this necessity for dual-mesh fidelity will inevitably become the standard procedure for all future articulated Object works.

---

### Meta-Review · Area_Chair_SLzd · 2026-01-06

**Summary:**

The paper proposes Real-IKEA, a simulation framework and dataset aimed at reducing the sim-to-real gap in contact-rich manipulation through high-fidelity collision meshes and visual rendering. While the reviewers universally acknowledged the solid engineering effort and the potential utility of the assets for the robotics community, the consensus is that the submission lacks the algorithmic or theoretical innovation expected at ICLR. The work is viewed as a valuable resource paper built by combining existing tools (CoACD, MuJoCo, standard rendering), rather than a fundamental advance in representation learning. Consequently, the work is deemed a better fit for a robotics-focused systems conference (e.g., ICRA, IROS, RSS) rather than ICLR.

**Reviewer Concerns:**

Addressed:

1）Visual Comparison Fairness: The authors re-ran experiments with standardized backgrounds to address concerns about unfair FID/EMD comparisons.

2）Asset Novelty: Clarified that assets are newly created and not merely subsets of existing datasets like GAPartManip.

3）Simulation Efficiency: Provided data on FPS with decomposed meshes.

Outstanding:

1）Technical Novelty: The consensus remains that the work relies heavily on combining existing off-the-shelf tools (MuJoCo, COACD) without significant algorithmic innovation relevant to ICLR.

2）Downstream Validation: The lack of a demonstrated sim-to-real policy transfer on a physical robot weakens the central claim of bridging the sim-to-real gap.

3）Methodology: Concerns persist regarding the manual specification/tuning of resistance parameters rather than data-driven grounding.

**Reviewer Scores:**

Reviewer WRPG (2): Likely remains 2-3 (Technical novelty concern persists).

Reviewer e7P3 (2): Likely remains 2-3 (Validation gap remains).

Reviewer miW6 (2): Likely remains 2-3 (Innovation concern persists).

Reviewer eEZQ (6): Likely remains 5-6 (Acknowledges dataset value but notes limited novelty).

---

### Decision · Program_Chairs · 2026-01-26

Reject